# Barriers to widespread adoption of telehealth from physicians' perspective: A survey in southern Iran

Mohammad Hosein Hayavi-haghighi[1,2], Niloofar Choobin[3], Jahanpour Alipour🆔[4]*

1 Associate Professor, Department of Health Information Technology, School of Allied Medical Sciences, Hormozgan University of Medical Sciences, Bandar Abbas, Iran, 2 Social Determinants in Health Promotion Research Center, Hormozgan Health Institute, Hormozgan University of Medical Sciences, Bandar Abbas, Iran, 3 Faculty of Para-medicine, Student Research Committee, Hormozgan University of Medical Sciences, Bandar Abbas, Iran, 4 Health Human Resources Research Center, School of Health Management and Information Sciences, Shiraz University of Medical Sciences, Shiraz, Iran

* jahanpour.alipour@gmail.com

## Abstract

### Introduction

The widespread use of telehealth in healthcare institutions depends on removing its barriers. This study examines the barriers to the widespread adoption of telehealth in the teaching hospitals of Hormozgan University of Medical Sciences (HUMS).

### Methods

A cross-sectional descriptive-analytical study was conducted in 2024 in three teaching hospitals. The study population included 239 physicians working at teaching hospitals of HUMS, all of them were included in the sample. Data collection was performed utilizing a questionnaire developed by the researchers. The analysis of the data involved both descriptive (mean and standard deviation) and analytical (Pearson's and Spearman's correlation coefficients) statistics.

### Results

A total of 169 physicians completed and returned the questionnaire. The mean score of personal, technical, behavioral, organizational, legal, clinical, and financial from the physician's perspective was 3.83±0.66, 3.81±0.56, 3.75±0.52, 3.66±0.53, 3.64±0.59, 3.59±0.55, and 3.54±0.67, respectively. The organizational factor had the highest correlation with others. Pearson's correlation test showed that there was a significant positive and fair correlation between the organizational factor and financial (r=0.524, P<0.01), clinical (r=0.399, P<0.01), technical (r=0.308, P<0.01), behavioral (r=0.321, P<0.01), and personal (r=0.307, P<0.01) factors.

**Data availability statement:** All relevant data are within the paper and its Supporting Information files.

**Funding:** The author(s) received no specific funding for this work.

**Competing interests:** The authors have declared that no competing interests exist.

## Conclusions

The correlation between factors demonstrates that the success of telehealth necessitates the consideration of a multitude of interdependent dimensions. Training physicians, improving the delivery of healthcare to patients, and developing and updating guidelines for telehealth services are potential solutions for eliminating the barriers.

## Introduction

The equitable and needs-based provision of care to all members of society has emerged as a pivotal concern for governments in recent decades. To achieve this goal, a variety of strategies are employed, with the development of telehealth representing one of the most effective approaches, particularly in the context of crises such as the COVID-19 pandemic [1,2]. The term 'telehealth' is used to describe a wide range of remote information and communication technologies that are employed to improve the delivery of healthcare services and optimize the performance of the healthcare system [3]. The field of telehealth has experienced a period of accelerated development in recent years, with the advent of the pandemic serving to accelerate its implementation and adoption [4]. Telehealth technologies have been applied across a diverse range of care domains, including assessment, diagnosis, treatment, therapy, follow-up, and monitoring [5–7]. The expansion of telehealth has been facilitated by advancements in information technology, such as the Internet of Things (IoT), Virtual Reality (VR), and Artificial Intelligence (AI) [8–10].

Despite the numerous advantages offered by telehealth, the field has encountered some barriers. It is evident that a number of these barriers pertain to the realm of policy-making and the provision of the necessary infrastructure (reimbursement, legal and technical policies) by the government [11–13]. In addition, a proportion of these barriers are associated with internal hospital issues (network reliability, change of clinical processes, data privacy and physician acceptance) [14,15]. Furthermore, a separate set of barriers relates to patients (health literacy, preferences and convenience) [16,17]. It is important to acknowledge that obstacles to the adoption of telehealth differ between high-income and low- or middle-income countries. While in high-income countries, the discourse predominantly focuses on reimbursement, quality of care, and eligibility [18,19], in low-income countries, challenges such as literacy, inadequate standards, and a lack of interest among patients and providers are more pronounced [20,21]. A comprehension of the impediments to telehealth adoption is instrumental in the implementation of efficacious strategies to optimize the capabilities of the health system in the future [22]. It is of particular importance to recognize and remove some of these challenges in environments with limited resources, such as in Iran [23,24].

Iran is confronted with significant challenges due to population dispersion, the presence of mountainous and inaccessible regions, the growth of the elderly population, and the lack of universal and equitable access to specialized medical facilities [25]. In this context, telehealth can be a valuable solution [26]. Furthermore, the

utilization of telehealth can facilitate expeditious disease diagnosis, the implementation of appropriate treatment strategies, the reduction of time wastage and the minimization of related costs [27]. Along with issues such as the inappropriate distribution of specialized staff and the concentration of facilities and equipment in large urban areas on the one hand [28] and the increase in chronic diseases and the desire to reduce the length of hospitalization and costs on the other, have increased the desire for home care of patients. This has in turn doubled the need to use telehealth [29].

Identifying existing barriers is a prerequisite for widespread adoption of telehealth. In Iran, Bijani et al. [30] identified organizational, professional, and equipment-related factors, while Hosseini et al. [31] identified the inappropriateness of virtual visits for physical examinations and lack of patient literacy as barriers to telehealth use.

Hospitals in Iran have not met the necessary standards for setting up a telehealth network [32]. The country has adopted minimum standards for telehealth, which are not as thorough as those in other countries. These standards do not improve performance or service quality [22]. Additionally, high costs and technical challenges related to data transmission have limited the use of telehealth services to very few places and situations [24].

Notwithstanding the global surge in telehealth adoption during the ongoing pandemic [33], evidence from Iran suggests that telehealth has been largely experimental and temporary, with initiatives largely spearheaded by individual physicians and constrained in scope [24,34]. Also The Ministry of Health has concentrated its efforts on utilizing telehealth for educational, executive, and psychological counselling purposes [35].

Despite advances in telecommunication technologies, evolving reimbursement mechanisms, and growing societal acceptance of telehealth, its full integration into healthcare systems depends on removing obstacles [36,37]. This necessitates a comprehensive and meticulous planning process. The Hormozgan University of Medical Sciences (HUMS) is responsible for the provision of health and treatment services to the residents of Hormozgan province.

The heat and humidity, and the dispersion of cities and villages in Hormozgan Province, pose challenges in providing healthcare services to patients. Telehealth has the potential to facilitate improved access to the services required by residents. It is imperative that senior health managers must implement telehealth policies in a timely manner. In order to facilitate this process, it is essential to first establish a consensus among all relevant stakeholders. In addition, it is crucial to identify and remove existing barriers to the implementation of telehealth. To date, no study has been conducted in Hormozgan to identify these barriers, despite the clear importance of the issue. Consequently, this study aims to examine the barriers to the widespread adoption of telehealth in the teaching hospitals of HUMS.

## Methods

### Study design and setting

This cross-sectional survey examined physicians' opinions regarding the potential barriers to the widespread adoption of telehealth in three teaching hospitals affiliated with Hormozgan University of Medical Sciences (HUMS). The hospitals involved were Shahid Mohammadi Hospital, a general facility with 400 beds; Shariati Hospital, which focuses on gynecology and obstetrics and has 139 beds; and Children's Hospital, dedicated to pediatrics with a capacity of 130 beds. The data were collected in September 2024 using an electronic questionnaire. This study was approved by the Ethics Committee of the Deputy of Research and Technology of Hormozgan University of Medical Sciences (No: IR.HUMS. REC.1402.352). The details of the ethical approval can be accessed at: https://ethics.research.ac.ir/EthicsProposalView. php?id=417496

### Study population and sample

The study population consisted of all 130 medical specialists and 109 general practitioners (GPs) working in teaching hospitals of HUMS. Because of population limitation, a sampling process was deemed unnecessary, and the entire population was utilized as the sample. The criteria for participant inclusion in the study were based on their willingness to engage and their prior experience with telehealth.

## Data collection tool

Although there are established models for assessing user acceptance of health information technologies, such as the Technology Acceptance Model (TAM) [38], the Unified Theory of Acceptance and Use of Technology (UTAUT) [39,40], and the DeLone and McLean information systems success model [41], the aim of these models is to analyze how different factors influence user behavior and their acceptance of these technologies [38–41]. This study specifically sought to identify barriers to the use of telehealth from the viewpoint of physicians; thus, technology acceptance models were not used to compile a comprehensive list of barriers and their relative importance. Thus, a questionnaire was developed with a specific focus on aligning with the study's objective. To create the questionnaire, a comprehensive search was conducted in the PubMed and Scopus databases to determine the barriers associated with the utilization of telehealth by practitioners. Subsequently, a search of SID was conducted to identify any Persian (Iranian) studies. A search was applied using the keywords "telehealth" and "barriers" along with their equivalents, spanning from 2000 to April 2023. The objective was to identify, categorize and report the barriers to the adoption of telehealth using a thematic analysis [3,42].

A preliminary questionnaire was subsequently developed, consisting of 64 questions organized into two sections. The initial section focuses on demographic information, including age, gender, level of education (specialist/general practitioner), and years of professional experience. The second section addresses the obstacles to the extensive adoption of telehealth in Bandar Abbas and contains 60 questions. This section presents seven categories of barriers to the adoption of telehealth, including organizational (n = 13), financial (n = 8), clinical (n = 13), technical (n = 11), and behavioral (n = 7), legal (n = 5), and personal barriers (n = 3). To rate each question, a five-point Likert scale (1 = strongly disagree, 2 = disagree, 3 = neutral, 4 = agree, and 5 = strongly agree) was used.

A content verification method was utilized to assess the draft questionnaire regarding its relevance, clarity, simplicity, and ambiguity. Two panels of experts were assembled, and the questionnaire was distributed to five individuals holding PhDs in health information management and three physicians, including two specialists and a general practitioner. The criteria for selecting experts are adequate practical experience in utilizing telehealth as well as a minimum of five years of relevant experience. The draft questionnaire was subsequently revised based on the recommendations provided by the expert panels.

Items that achieved a content validity index (I-CVI) of ≥ 0.75 were retained, those with an I-CVI of 0.70–0.75 were modified, and items with an I-CVI of ≤ 0.70 were removed. Following the feedback, one clinical question was eliminated. Consequently, the final version of the questionnaire included 12 questions addressing clinical barriers, contributing to a total of 63 questions. Given that there were eight experts, the validity of the questions was considered acceptable if the content validity ratio (CVR) was greater than 0.75 and the content validity index (CVI) was greater than 0.75 [43]. Additionally, the content validity index for each of the seven of barriers in the questionnaire was greater than 80 and was approved.

To assess the reliability of tool, the questionnaire was given to 20 participants over a 10-day period and validated using Cronbach's alpha correlation test, which yielded a value of 0.91. Moreover, the Cronbach's alpha for all subscales exceeded 0.7, suggesting that they were satisfactory and were approved. The more details of the validated questionnaire can be seen in S1 Appendix 1-Questionnaire.

Data was collected using electronic self-administered questionnaire. After obtaining code of ethics the researchers send the link of questionnaire to the physicians. After clicking on the questionnaire's link, physicians were presented with a page explaining the purpose of the study.

The participants' consent to partake in the study was obtained in written and electronic form. A box was located at the bottom of the page, requesting consent from the participants for their involvement in the study. After ticking the box, the questionnaire was made available to the physicians. To ensure that more physicians would complete it, a reminder message was sent to participants twice, with an interval of one week between each reminder.

## Data analysis

Collected data were entered into SPSS software version 22. The demographic data was displayed using descriptive statistics, including frequency and percentage. The data on barriers to the use of telehealth were analyzed by mean and Standard Deviation (SD). Furthermore, the mean score of the factors of the examined challenges was used to explain the importance of the challenges of telehealth implementation and use from physicians' perspective so that a higher mean score was considered more serious. Additionally, we applied Pearson correlation tests to examine the connections between the main variables of personal, technical, behavioral, organizational, legal, clinical, and financial, as well as age and work experience in relation to these variables. We also used Spearman correlation test to investigate the relationship between gender and the other variables. In all analyses, we regarded a P-value less than 0.05 to be significant.

## Results

### Participants' demographic characteristics

Of the total number of participants (n = 239), 169 completed the electronic questionnaire. The participants' mean age and work experience were 37.38 ± 8.059 and 9.56 ± 7.421 years, respectively. Most participants were female (54.4%), had 10 or fewer years of work experience (65.12%), and were specialist (52.1%). Besides, the highest frequency of their age group was in the 34–43 subcategory (Table 1).

### The mean score and SD of the surveyed factors

Physicians identified personal and technical factors as the main barriers to telehealth, and clinical and financial factors as the least important (Fig 1).

### Correlation of evaluated factors

The organizational factor had the highest correlation with other effective factors in the adoption of telehealth. Pearson's correlation test showed that there was a significant positive and fair correlation between the organizational factor and financial (r = 0.524, P < 0.01), clinical (r = 0.399, P < 0.01), technical (r = 0.308, P < 0.01), behavioral (r = 0.321, P < 0.01), and personal (r = 0.307, P < 0.01) factors; between financial factor and technical factor (r = 0.326, P < 0.01); between clinical factor and technical (r = 0.321, P < 0.01) and behavioral (r = 0.403, P < 0.01) factors; between technical factor and behavioral (r = 0.400, P < 0.01), legal (r = 0.308, P < 0.01) and behavioral (r = 0.451, P < 0.01) factors, between behavioral factor and legal (r = 0.475, P < 0.01) and personal (r = 0.421, P < 0.01) factors; and between legal factor with personal factor (r = 0.312, P < 0.01).

**Table 1. Demographic Characteristics (n = 169).**

| Category | Subcategory | N (%) |
|---|---|---|
| Gender | Female | 92 (54.4) |
| | Male | 77 (45.6) |
| Age (year) | 24-33 | 64 (37.9) |
| | 34-43 | 68 (40.2) |
| | 44-53 | 29 (17.2) |
| | 54-63 | 8 (4.7) |
| Work experience (year) | ≤ 10 | 110 (65.1) |
| | 11-20 | 37 (21.9) |
| | 21-30 | 22 (13) |
| Education | General practitioner (GP) | 81 (47.9) |
| | Specialist | 88 (52.1) |

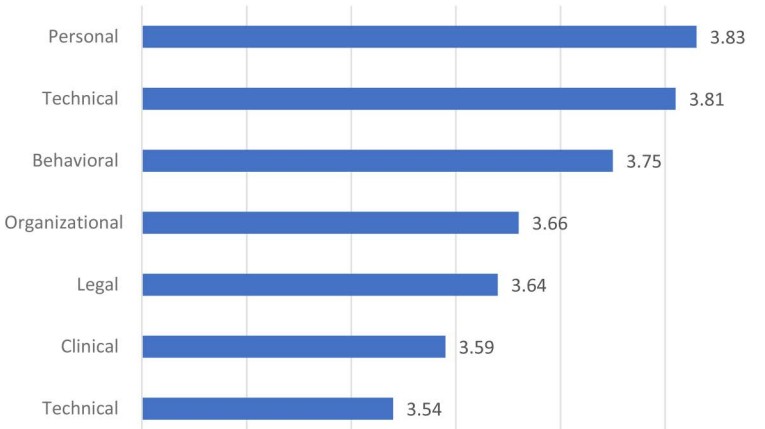

**Fig 1. The mean of various investigated factors from physicians' perspective.**

There is a significant positive and poor correlation between the financial factor and clinical (r = 0.292, P < 0.01), behavioral (r = 0.235, P < 0.01), legal (r = 0.190, P < 0.05) and personal (r = 0.215, P < 0.01) factors; between the clinical factor and legal (r = 0.275, P < 0.01) and personal (r = 0.231, P < 0.01) factors; and between the organizational and legal factors (r = 0.213, P < 0.01). Age (r = 0.162, P < 0.05) and work experience (r = 0.184, P < 0.05) only had a significant positive but poor correlation with the personal factor. As well as, there was a significant positive and strong correlation between age and work experience (r = 0.886, P < 0.01). **Table 2**

Furthermore, Spearman's correlation coefficient test showed that there was no significant relationship between participants' gender and different variables (P > 0.05).

## Discussion

To the best of our knowledge, this cross-sectional study is the first to comprehensively explore the barriers to telehealth implementation from the perspective of physicians in Iran. telehealthThis survey also highlights important but complex barriers to the implementation of telehealth in Iran.

Our results show that personal factors, such as limited access, lack of digital health literacy, and insufficient technical staff, were the main barriers to telehealth implementation. Wasi Abbas et al. [44] found a lack of patient digital literacy

**Table 2. The correlation of evaluated factors from the physicians' perspective.**

| Variables | | 1 | 2 | 3 | 4 | 5 | 6 | 7 | 8 | 9 |
|---|---|---|---|---|---|---|---|---|---|---|
| 1 | Organizational factor | 1 | | | | | | | | |
| 2 | Financial factor | .524** | 1 | | | | | | | |
| 3 | Clinical factor | .399** | .292** | 1 | | | | | | |
| 4 | Technical factor | .308** | .326** | .321** | 1 | | | | | |
| 5 | Behavioral factor | .321** | .235** | .403** | .400** | 1 | | | | |
| 6 | Legal factor | .213** | .190* | .275** | .308** | .475** | 1 | | | |
| 7 | Personal factor | .307** | .215** | .231** | .451** | .421** | .312** | 1 | | |
| 8 | age | .065 | .107 | .035 | .040 | −.015 | .023 | .162* | 1 | |
| 9 | experience | −.042 | −.015 | .036 | .007 | −.028 | −.017 | .184* | .886** | 1 |

* P- value < 0.05

** P- value < 0.01

to be a key barrier to telehealth adoption in Pakistan, while Kubota et al. highlighted this as an issue in Japan [45]. To remove personal barriers to telehealth, improve digital health literacy levels, and provide more medical and technical staff in rural and remote areas is essential.

The second most common barrier to telehealth from a physicians' perspective is technical. Lack of suitable infrastructure and internet connection failure are the most important sub-factors of technical barriers. In line with our findings, technical difficulties were the first and second biggest barriers to the use of telehealth in Saudi Arabia and Japan respectively [45,46]. Amiri et al also identified infrastructure and technical difficulties as the main barriers to the use of technology in Iran [47]. Investment in infrastructure and telehealth could help overcome existing barriers.

Previous studies have shown that end-user (physician and patient) reluctance to use telehealth and cultural difficulties (user preference for face-to-face interaction) are critical barriers to technology adoption [48,49]. Our results indicate that behavioral factors are the third most common barrier to telehealth use from the perspective of physicians. Furthermore, cultural beliefs and low end-user knowledge about telehealth technology and its benefits are among the most important behavioral issues in this study. The study by Habib et al. also identified physician resistance as one of the major barriers to the use of telehealth [46]. The integration of medical practitioners into the implementation phases of telehealth, coupled with the effective dissemination of its advantages to all relevant parties, is poised to significantly mitigate the impact of behavioral impediments.

The study identified organizational barriers as the fourth most common barrier telehealth and the improper end-user training, inadequate support staff, and the lack of documented program at different levels of the country for the implementation of telehealth as the most important organizational factors identified by the physicians.

The term 'organizational aspects' is generally understood to refer to those elements which manage the processes of health service delivery and affect the manner in which all institutions and individuals involved interact. Such elements may include the Ministry of Health, senior university and hospital management, physicians, patients, and technical and support staff. It is therefore asserted that the removal of organizational barriers requires proper planning, adequate training, the provision of relevant staff, and the strengthening of communication with all stakeholders [46].

Previous studies highlighted the legal barriers such as patients' security, confidentiality, and privacy issues as obstacles against successful implementation of telehealth [44,45,48]. Our results show that physicians identified the legal factor as the fifth most common barrier to the widespread use of telehealth. In addition, concerns about forensic medicine (medical negligence and malpractice), patient safety and confidentiality, difficulties in obtaining informed consent from patients, and patient privacy issues were among the top legal barriers identified by physicians. Adopting international standards such as ISO/IEC 27001 or a national HIPPA-like standard for patient data security, training physicians to obtain informed consent for telehealth, and improving the level of care for patients are solutions to reduce legal barriers.

Our results show that clinical barriers are the sixth most common barriers to implementing telehealth from physicians' perspective. Furthermore, the lack of access to clinical patient care protocols and limitations in physical examination were identified as the most important clinical barriers. It is important to acknowledge that clinical barriers can have an inhibitory effect on the process and quality of care delivery. In line with the findings of this study, similar studies have identified the lack of physical examination and the possibility of misinterpretation of diagnoses as the main barriers [3,50–55]. Wearable devices may be a promising solution to facilitate some virtual examinations and reduce these barriers [56].

The results of this study indicate that physicians cited financial barriers as the last barrier to implementing telehealth. However, reimbursement concerns by insurance companies and insufficient budget allocation at different levels (national, local and organizational) for the implementation of telehealth were identified as the most important financial barriers. Previous studies have also highlighted reimbursement issues as an important barrier to the use of telehealth [42,48]. During the pandemic, many insurance companies have shown great flexibility by adapting and simplifying their policies, which has led to the welcome development of reimbursement rates for telehealth services that are now comparable to those for face-to-face visits [12,57]. This initiative has undoubtedly contributed to the significant growth in telehealth use during and

after the pandemic. Insurance companies in Iran may also find it beneficial to consider the experiences of their peers and consider ways to develop the necessary regulations and facilitate reimbursement for telehealth services.

The results of Pearson's correlation test showed that there is a significant positive and fair relationship between organizational factor and financial, clinical, technical, behavioral, and personal factors; between financial and technical factors; between clinical and technical and behavioral factors; between technical and behavioral, legal and behavioral factors, between behavioral and legal and personal factors; and between legal with personal factors ($P < 0.05$). Furthermore, there is a significant positive and poor correlation between the financial and clinical, behavioral, legal and personal factors; between the clinical and legal and personal factors; and between the organizational and legal factors.

### Study strengths and limitations

In this research, a thorough and precise statistical analysis, adhering to robust research practices, has been employed to offer scientific understanding of the obstacles preventing physicians from widely adopting telehealth. This opens up opportunities for investigating similar situations. As a result, the insights gained can assist policy-makers and managers in formulating critical policies and strategies to address the barriers to telehealth adoption by physicians in the given context.

However, there are limitations to our study. Firstly, this study exclusively examined the perspectives of physicians. In order to successfully identify barriers to the adoption of telehealth, it is essential to involve all stakeholders, including patients, all care providers, senior management of care organizations, and health ministry officials. It is recommended that future studies examine the perspectives of other stakeholders.

Secondly, although most physicians participated in this study, we were unable to obtain responses and perceptions from some of them regarding their use of telehealth services. Therefore, our results may be affected by non-response bias.

Thirdly, this study was conducted in a city in south of Iran (Bandar Abbas) and therefore its results cannot be generalized to the whole country of Iran. Fourthly, this study was conducted using physician self-reporting and may therefore be subject to self-reporting bias.

Finally, this study is quantitative and cross-sectional, meaning it has limits in showing cause-and-effect relationships. The self-reported data could also be influenced by social desirability bias, where participants may answer in a way, they think looks good. Future studies could use qualitative or mixed methods to better understand the barriers to telehealth use in Iran.

### Conclusion

In their analysis, physicians identified the most significant barriers to the widespread adoption of telehealth, which they classified into seven interrelated categories: personal, technical, behavioral, organizational, legal, clinical, and financial. The correlation between these factors demonstrates that the successful implementation of telehealth necessitates the consideration of a multitude of interdependent dimensions. Training physicians, improving the delivery of healthcare to patients, utilizing national and international information protection safeguards to guarantee security, confidentiality, and privacy, and developing and updating reimbursement regulations and guidelines for telehealth healthcare services are potential solutions for eliminating the aforementioned barriers. In order to successfully implement telehealth in Iran, health planners must adopt a coherent approach in partnership with other health-related entities, including payers, IT developers, professional associations of physicians and paramedics, and patient rights organizations.

### Supporting information

**S1 Appendix. Questionnaire.docx.**
(DOCX)

## Acknowledgments

The authors thankful to the physicians of educational hospital of Hormozgan University of Medical Sciences who participated in this study for sharing their valuable experiences.

## Author contributions

**Conceptualization:** Mohammad Hosein Hayavi-haghighi, Niloofar Choobin, Jahanpour Alipour.

**Data curation:** Mohammad Hosein Hayavi-haghighi, Niloofar Choobin, Jahanpour Alipour.

**Formal analysis:** Jahanpour Alipour.

**Methodology:** Mohammad Hosein Hayavi-haghighi, Jahanpour Alipour.

**Writing – original draft:** Mohammad Hosein Hayavi-haghighi, Niloofar Choobin, Jahanpour Alipour.

**Writing – review & editing:** Mohammad Hosein Hayavi-haghighi, Niloofar Choobin, Jahanpour Alipour.

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
