## [Decision Letter · Decision Letter 0]

Dear Dr. Alipour,

Thank you for submitting your manuscript to PLOS ONE. After careful consideration, we feel that it has merit but does not fully meet PLOS ONE’s publication criteria as it currently stands. Therefore, we invite you to submit a revised version of the manuscript that addresses the points raised during the review process.

We look forward to receiving your revised manuscript.

Kind regards,

Marsa Gholamzadeh, PhD

Academic Editor

PLOS ONE

Journal Requirements:

4. We are unable to open your Supporting Information file telemedicine.sav. Please kindly revise as necessary and re-upload.

Reviewers' comments:

Reviewer's Responses to Questions

**Comments to the Author**

1. Is the manuscript technically sound, and do the data support the conclusions?

Reviewer #1: Partly

Reviewer #2: Yes

2. Has the statistical analysis been performed appropriately and rigorously?

Reviewer #1: No

Reviewer #2: Yes

3. Have the authors made all data underlying the findings in their manuscript fully available?

Reviewer #1: No

Reviewer #2: Yes

4. Is the manuscript presented in an intelligible fashion and written in standard English?

Reviewer #1: Yes

Reviewer #2: Yes

Reviewer #1: Well done to conceive of this study and the for writing up the study.

You have stated that the goal of the study is not clear as to "examine" barriers to telehealth adoption at a particular regional setting in Iran, but it would be important to summarise what is already known about some of the barriers to adoption of telehealth worldwide and in Iran itself, and then localise it to the hospital setting. Just because some barriers are identified at the provider level does not mean that addressing them will increase adoption of telehealth by the providers or patients. The views of patients or who will use telehealth would also be important, and is missing in the manuscript. The results of the study are not generalisable and cannot be applied to other situations or indeed it is doubtful if the results can be used in the wider context of Iran for that matter.

In the methods section, you have not indicated any theory on the basis of which you drafted the questionnaire. For example, "Technology Acceptance Model" (TAM) is widely used for investigating adoption of technology and identification of barriers, and the questionnaires and instruments have been validated. Why such models were not considered in this case needs to be explained in the Methods section. In the methods section, you have only provided item-level content validity score, although you have seven subscales. You need to provide scale level content validity index as well. What the Cronbach's alpha measured in this context is not clear, as Cronbach's alpha is used for a scale but as you have not mentioned development of a scale as such, you may want to explain the scale development process in details. In the Methods section, you wrote you sent an electronic questionnaire to the doctors who would have to access and answer them. How did you make sure that the doctor completed the questionnaire, not the doctor's son or daughter or the doctor's assistant or a student or someone other than the doctor completed the questionnaire? How did you guard against that possibility?

In the Results section you have written 70% of the target audience responded to the questionnaire. What happened to the remaining 30% or 70 doctors. Why did they not participate? This information is important for understanding the significance of the results. Secondly, you have mentioned that there were 7 factors and 63 questions with different number of questions per factor. Now, you have reported factor level correlations, but these correlations were driven by item level correlations. it is important to report the factor structure first, before factor level reporting. It seems that you may have conducted confirmatory factor analyses but you have not specified which items did which factor load on, and therefore this factor level correlations are uninterpretable. If you conduct a formal structural equation model with a confirmatory factor analysis of the items, then it might be possible to interpret the results of the findings of your study.

Reviewer #2: Corrections, Comments, and Suggestions

Below is an assessment of different sections of the manuscript:

1. Introduction

Strengths:

Provides a clear background on the importance of telemedicine.

Discusses the challenges specific to Iran.

Suggested Corrections & Improvements:

The transition to the research objective could be clearer. The last paragraph should explicitly state how the study addresses the gap in research.

The importance of the study should be framed in a way that highlights how it contributes to policy recommendations or future implementation strategies.

2. Methods

Strengths:

Clearly describes the study design, setting, and population.

Provides details on questionnaire development, validation, and reliability testing.

Suggested Corrections & Improvements:

The ethical approval section should explicitly mention whether written informed consent was obtained from participants.

The statistical analysis section could benefit from a more detailed explanation of why specific tests (Pearson’s & Spearman’s correlation) were used and how they align with the study objectives.

3. Results

Strengths:

Well-structured, with clear presentation of findings.

Use of correlation analysis adds depth to understanding relationships among barriers.

Suggested Corrections & Improvements:

Consider adding a figure or chart to visually represent the correlations between barriers for better clarity.

The demographic table (Table 1) should be formatted to ensure all categories are properly aligned for readability.

Some percentages in Table 1 seem inconsistent or misplaced (e.g., "11-20 years of experience" percentage appears incorrect).

4. Discussion

Strengths:

Compares findings with prior studies from different countries.

Provides practical recommendations for addressing barriers.

Suggested Corrections & Improvements:

Some points are repetitive (e.g., legal and financial barriers are discussed twice in slightly different ways). Consolidation would improve clarity.

While discussing financial barriers, the role of insurance companies in telemedicine adoption could be further elaborated.

The study implications should be expanded to include how policymakers and hospital administrators can implement change based on these findings.

5. Study Strengths and Limitations

Strengths:

Recognizes methodological rigor and policy implications.

Suggested Corrections & Improvements:

The limitations should discuss potential bias (e.g., self-reporting bias due to the questionnaire-based nature of data collection).

A future recommendation for qualitative interviews with physicians could add depth to understanding the barriers.

6. Conclusion

Strengths:

Summarizes key findings effectively.

Suggests actionable solutions for overcoming telemedicine barriers.

Suggested Corrections & Improvements:

The last sentence could call for further research on solutions rather than just stating that future studies should focus on removing obstacles.

Consider mentioning specific policy changes that could enhance telemedicine adoption in Iran.

Overall Assessment

Strengths:

Well-organized and methodologically sound.

Identifies multiple dimensions of telemedicine barriers.

Provides actionable recommendations.

Areas for Improvement:

Improve clarity in transitions (especially in the Introduction and Conclusion).

Consolidate redundant discussion points.

Add visual elements (charts or tables) to enhance readability.

Address minor formatting and statistical explanation gaps in the Methods and Results sections.

**Do you want your identity to be public for this peer review?** For information about this choice, including consent withdrawal, please see our Privacy Policy

Reviewer #1: **Yes: ** Arindam Basu

Reviewer #2: **Yes: ** Mekides Molla Reda

---

## [Author Response · Author response to Decision Letter 1]

3 Mar 2025

Dear editor and reviewers of PLOS ONE

We would like to express our sincere gratitude for the scientific and constructive approach adopted during the review process. We have invested a considerable amount of time and effort to address the comments made, and it is our belief that the manuscript has undergone significant enhancement in quality as a result.

The following table presents the responses to the comments made by the esteemed editor and reviewers.

Editor and reviewers’ comments Response to Comments

Editor comments

1. Please ensure that your manuscript meets PLOS ONE's style requirements, including those for file naming. Minor corrections were made according to the PLOS ONE styles.

2. Please provide additional details regarding participant consent. In the ethics statement in the Methods and online submission information, please ensure that you have specified (1) whether consent was informed and (2) what type you obtained (for instance, written or verbal, and if verbal, how it was documented and witnessed). If your study included minors, state whether you obtained consent from parents or guardians. If the need for consent was waived by the ethics committee, please include this information. Necessary explanations about written informed consent are provided in the ethics statement.

3. Your ethics statement should only appear in the Methods section of your manuscript. If your ethics statement is written in any section besides the Methods, please delete it from any other section. The method section encompassed all matters pertaining to ethics.

4. We are unable to open your Supporting Information file telemedicine.sav. Please kindly revise as necessary and re-upload. All the required data is within the article. There is no need for the referenced appendix.

Please include captions for your Supporting Information files at the end of your manuscript, and update any in-text citations to match accordingly. Please see our Supporting Information guidelines for more information: Minor corrections were made according to the PLOS ONE styles.

Reviewer 1

You have stated that the goal of the study is not clear as to "examine" barriers to telehealth adoption at a particular regional setting in Iran, but it would be important to summarise what is already known about some of the barriers to adoption of telehealth worldwide and in Iran itself, and then localise it to the hospital setting. Just because some barriers are identified at the provider level does not mean that addressing them will increase adoption of telehealth by the providers or patients. In a new paragraph on the first page of the introduction, the barriers to telehealth implementation were categorised into three levels. In addition, in the second paragraph on the second page of the introduction, reference was made to some of the barriers reported in Iran.

The views of patients or who will use telehealth would also be important, and is missing in the manuscript. The restriction of the study's scope to the perspective of physicians served to compound the research's limitations.

The results of the study are not generalisable and cannot be applied to other situations or indeed it is doubtful if the results can be used in the wider context of Iran for that matter. We have also mentioned this limitation in the study limitations section.

In the methods section, you have not indicated any theory on the basis of which you drafted the questionnaire. For example, "Technology Acceptance Model" (TAM) is widely used for investigating adoption of technology and identification of barriers, and the questionnaires and instruments have been validated. Why such models were not considered in this case needs to be explained in the Methods section.

In the methods section, you have only provided item-level content validity score, although you have seven subscales. You need to provide scale level content validity index as well.

What the Cronbach's alpha measured in this context is not clear, as Cronbach's alpha is used for a scale but as you have not mentioned development of a scale as such, you may want to explain the scale development process in details.

In the Methods section, you wrote you sent an electronic questionnaire to the doctors who would have to access and answer them. How did you make sure that the doctor completed the questionnaire, not the doctor's son or daughter or the doctor's assistant or a student or someone other than the doctor completed the questionnaire? How did you guard against that possibility? We agree with the reviewer that there are multiple credible models available for understanding user acceptance of technologies, such as various versions of UTAUT and TAM, among others. These well-established models serve to examine how different aspects affect user behavior and their acceptance of these technologies. In our study, our focus was exclusively on the barriers to telehealth technology adoption from the perspective of physicians, which is why we chose not to utilize the existing models. As a result, a clarification has been added in the Methods section.

Along with calculating the content validity score for the items, we also assessed the content validity of the subscales. We had erroneously assumed that reporting validation of the questions automatically meant all subscales were validated as well. To clarify this point, we will include an additional sentence in the Methods section.

We agree with the reviewer and believe that the Cronbach's alpha value for the complete questionnaire does not adequately represent the condition of the subscales. As such, we apologize for any unintentional confusion. The preceding sentence has been modified and rewritten to eliminate the ambiguity.

Concerning whether the physician filled out the questionnaire personally or if it was filled out by someone else, it remains uncertain who provided the responses for both in-person visits and paper questionnaires. Regrettably, there was no resolution to this issue. This represents one of the limitations inherent in this kind of study and we mention it in the limitation section of the manuscript.

In the Results section you have written 70% of the target audience responded to the questionnaire. What happened to the remaining 30% or 70 doctors. Why did they not participate? This information is important for understanding the significance of the results. Secondly, you have mentioned that there were 7 factors and 63 questions with different number of questions per factor. Now, you have reported factor level correlations, but these correlations were driven by item level correlations. it is important to report the factor structure first, before factor level reporting. It seems that you may have conducted confirmatory factor analyses but you have not specified which items did which factor load on, and therefore this factor level correlations are uninterpreted. If you conduct a formal structural equation model with a confirmatory factor analysis of the items, then it might be possible to interpret the results of the findings of your study. Despite repeated follow-up, 70 physicians were unwilling to participate in the study (due to unwillingness to participate in the study and some due to busy work schedules). We have mentioned this in the limitations section.

Thank you for your scientific comment. In this study, confirmatory factor analyses were not the aim of our study. We apologize for using the word factor and causing ambiguity. We extracted the variables and related items based on the review of previous studies and only examined these items and also evaluated the correlation between the variables. Therefore, our goal was not a formal structural equation model

Reviewer 2

Introduction

The transition to the research objective could be clearer. The last paragraph should explicitly state how the study addresses the gap in research.

The importance of the study should be framed in a way that highlights how it contributes to policy recommendations or future implementation strategies. In the final paragraph of the introduction, further clarifications were provided to elucidate the purpose of the study.

In addition, further details were incorporated into the final paragraph in order to facilitate a more comprehensive understanding of the potential future impacts on managers and policymakers.

Methods

The ethical approval section should explicitly mention whether written informed consent was obtained from participants.

The statistical analysis section could benefit from a more detailed explanation of why specific tests (Pearson’s & Spearman’s correlation) were used and how they align with the study objectives. A section pertaining to ethical principles has been incorporated, wherein it is explicitly stated that written consent was obtained prior to the commencement of any research activities.

We updated and reformulated the Data Analysis subsection of the Methods section to enhance clarity.

Results

Consider adding a figure or chart to visually represent the correlations between barriers for better clarity.

The demographic table (Table 1) should be formatted to ensure all categories are properly aligned for readability.

Some percentages in Table 1 seem inconsistent or misplaced (e.g., "11-20 years of experience" percentage appears incorrect). We agree with the scientific feedback provided by the reviewer, which we reviewed with our statistics specialist. In this article, due to the extensive number of variables assessed and the resulting multitude of correlations (21 correlations just among the primary variables), it is virtually unfeasible to visually represent the correlations, as creating a graph for paired correlation types is impractical for displaying the data within the article. To enhance the clarity of the results presented in the paper, we transformed the data from Table 2 into figure 1.

The issues with Table 1 have been addressed, and the educational levels of the participants have also been included in the table.

Discussion

Some points are repetitive (e.g., legal and financial barriers are discussed twice in slightly different ways). Consolidation would improve clarity.

While discussing financial barriers, the role of insurance companies in telemedicine adoption could be further elaborated.

The study implications should be expanded to include how policymakers and hospital administrators can implement change based on these findings. The duplicate paragraph has been removed.

The role of insurance companies was also addressed in the discussion of financial barriers.

Solutions for managers and policymakers are mentioned both in the discussion of each category of obstacles and in the conclusion section.

Study Strengths and Limitations

The limitations should discuss potential bias (e.g., self-reporting bias due to the questionnaire-based nature of data collection).

A future recommendation for qualitative interviews with physicians could add depth to understanding the barriers. Self-reporting bias was added to the list of limitations

A qualitative approach has been suggested for future studies.

Conclusion

The last sentence could call for further research on solutions rather than just stating that future studies should focus on removing obstacles.

Consider mentioning specific policy changes that could enhance telemedicine adoption in Iran. The necessary corrections were duly implemented.

Overall Assessment

Improve clarity in transitions (especially in the Introduction and Conclusion).

Consolidate redundant discussion points.

Add visual elements (charts or tables) to enhance readability.

Address minor formatting and statistical explanation gaps in the Methods and Results sections. The introduction and method underwent necessary changes.

Duplicate content has been removed.

We agree with the scientific feedback provided by the reviewer, which we reviewed with our statistics specialist. In this article, due to the extensive number of variables assessed and the resulting multitude of correlations (21 correlations just among the primary variables), it is virtually unfeasible to visually represent the correlations, as creating a graph for paired correlation types is impractical for displaying the data within the article. To enhance the clarity of the results presented in the paper, we transformed the data from Table 2 into figure 1.

Minor formatting and statistical explanation revised and updated in the method section.

---

## [Decision Letter · Decision Letter 1]

Dear Dr. Alipour,

Thank you for submitting your manuscript to PLOS ONE. After careful consideration, we feel that it has merit but does not fully meet PLOS ONE’s publication criteria as it currently stands. Therefore, we invite you to submit a revised version of the manuscript that addresses the points raised during the review process.

We look forward to receiving your revised manuscript.

Kind regards,

Marsa Gholamzadeh, PhD

Academic Editor

PLOS ONE

Journal Requirements:

Reviewers' comments:

Reviewer's Responses to Questions

**Comments to the Author**

Reviewer #3: (No Response)

Reviewer #4: All comments have been addressed

2. Is the manuscript technically sound, and do the data support the conclusions?

Reviewer #3: Yes

Reviewer #4: Partly

3. Has the statistical analysis been performed appropriately and rigorously?

Reviewer #3: Yes

Reviewer #4: I Don't Know

4. Have the authors made all data underlying the findings in their manuscript fully available?

Reviewer #3: Yes

Reviewer #4: Yes

5. Is the manuscript presented in an intelligible fashion and written in standard English?

Reviewer #3: Yes

Reviewer #4: Yes

Reviewer #3: Firstly, on a positive note, I must say the language used is quite adequate and easy to read.

I can see that you have addressed all the comments made by reviewers, mostly in a well-structured manner. Nevertheless, I must make some comments:

- Firstly, referencing should happen at the end of phrases, not in the middle of the text. This is easily corrected.

- Secondly, the Editor comments on the availability of the appendix data, your answer was: ”All the required data is within the article. There is no need for the referenced appendix.”. In all honesty, I do not think this is a good enough answer; if there was an appendix, it should be made available. In my opinion, knowing that a questionnaire was used, it should be available to readers.

- Thirdly, I don’t think “adoption” should be used as a keyword for your work. I noticed the other Reviewers did not comment on it, but I suggest you evaluate the need for this word.

- Fourth: in my opinion, the paragraph “Data was collected using electronic self-administered questionnaire. After (…) a reminder message was sent to participants twice, with an interval of one week between each reminder”, should be in the “Data col-lection” part, not in ethics

- Lastly, I think your English is very good and the paper is well structured; however, some segments are long and appear repetitive. I believe you could summarize those segments better and you work could be more concise. Some examples are:,

o “In other to establish the foundations for the widespread adoption of telehealth in Iran, it is first necessary to (…), have resulted in the utilization of telehealth capabilities being largely confined tom a narrow range of locations and contexts [24].” >> could be more succinct.

o “Despite the ongoing advancement of telecommunication technologies, evolv-ing reimbursement mechanisms and an increasing societal (…) and removal of inherent obstacles”.

o “The Hormozgan province has a hot and humid climate (…). Consequently, this study aims to examine the barriers to the widespread adoption of telehealth in the teaching hospitals of HUMS” >> could be more succinct.

I hope the authors take the commentaries above as constructive observations that aim at help-ing them with the changes that might be made for the paper to be published.

Reviewer #4: Additional comments:

1. The authors didnot explicitly list teh weaknesses of the designs, the barriers would have been assessed better with a mixed methods design- the qualitative methods would have given indepth responses to the barriers.

2. There is no mention on Non- response bias, there were over 200 participants and only 160 responded to the survey, did the authors assess why the others didnot respond.

**Do you want your identity to be public for this peer review?** For information about this choice, including consent withdrawal, please see our Privacy Policy

Reviewer #3: No

Reviewer #4: **Yes: ** Dr. Agnes Bwanika Naggirinya

---

## [Author Response · Author response to Decision Letter 2]

22 May 2025

Dear editor and reviewers of PLOS ONE

We would like to express our sincere gratitude for the scientific and constructive approach adopted during the review process. We have invested a considerable amount of time and effort to address the comments made, and it is our belief that the manuscript has undergone significant enhancement in quality as a result.

The following table presents the responses to the comments made by the esteemed editor and reviewers.

Editor and reviewers’ comments Response to Comments

Reviewer #3

- Firstly, referencing should happen at the end of phrases, not in the middle of the text. This is easily corrected. Response: Most of the references cited have been revised. A few references have been added in the middle of text to better reflect the referred subject.

-Secondly, the Editor comments on the availability of the appendix data, your answer was: ”All the required data is within the article. There is no need for the referenced appendix.”. In all honesty, I do not think this is a good enough answer; if there was an appendix, it should be made available. In my opinion, knowing that a questionnaire was used, it should be available to readers. Response: We agree with the reviewer and the questionnaire used is attached accordingly.

- Thirdly, I don’t think “adoption” should be used as a keyword for your work. I noticed the other Reviewers did not comment on it, but I suggest you evaluate the need for this word. Response: We agree with the reviewer and we delete the adoption from keywords section.

- Fourth: in my opinion, the paragraph “Data was collected using electronic self-administered questionnaire. After (…) a reminder message was sent to participants twice, with an interval of one week between each reminder”, should be in the “Data col-lection” part, not in ethics Response: We thank the reviewer for his/her careful comments. The cited sentences have been moved to the relevant sections.

Lastly, I think your English is very good and the paper is well structured; however, some segments are long and appear repetitive. I believe you could summarize those segments better and you work could be more concise. Response: The mentioned sections are summarized as follows:

In order to establish the foundations for the widespread adoption of telehealth in Iran, it is first necessary to identify the barriers that currently exist. A study was conducted that reported on various barriers to the use of telecardiology, including organizational, professional and equipment-related issues [30]. Another study reported on the opinions of patients regarding virtual examinations, the inappropriateness of virtual visits for physical examinations, and the lack of patient literacy as the main barriers to tele-urology in Iran [31]. Nevertheless, hospitals in Iran have not yet met the standards required for the implementation of the telehealth network [32]. Furthermore, Iran has adopted the minimum standards for telehealth, which are less comprehensive than those of other countries. These standards are inadequate for enhancing performance and service quality [22]. Furthermore, the considerable financial and data volume costs, in addition to the technical challenges associated with their transmission, have resulted in the utilization of telehealth capabilities being largely confined to a narrow range of locations and contexts [24]. Identifying existing barriers is a prerequisite for widespread adoption of telehealth. In Iran, Bijani et al. [30] identified organizational, professional, and equipment-related factors, while Hosseini et al. [31] identified the inappropriateness of virtual visits for physical examinations and lack of patient literacy as barriers to telehealth use.

Hospitals in Iran have not met the necessary standards for setting up a telehealth network [32]. The country has adopted minimum standards for telehealth, which are not as thorough as those in other countries. These standards do not improve performance or service quality [22]. Additionally, high costs and technical challenges related to data transmission have limited the use of telehealth services to very few places and situations [24].

Despite the ongoing advancement of telecommunication technologies, evolving reimbursement mechanisms and an increasing societal inclination towards telehealth adoption (27, 28), the extensive integration of telehealth into healthcare systems is contingent upon the identification and removal of inherent obstacles. Despite advances in telecommunication technologies, evolving reimbursement mechanisms, and growing societal acceptance of telehealth, its full integration into healthcare systems depends on removing obstacles [36, 37].

The Hormozgan province's hot, humid climate, high latitude, and scattered islands in the Persian Gulf pose significant challenges to providing health services. The heat and humidity, and the dispersion of cities and villages in Hormozgan Province, pose challenges in providing healthcare services to patients.

Reviewer #4

1. The authors didnot explicitly list the weaknesses of the designs, the barriers would have been assessed better with a mixed methods design- the qualitative methods would have given indepth responses to the barriers. We thank the reviewer for her scientific and constructive comments and for taking the time to review the article.

Response: We agree with the reviewer and accordingly the following sentence has been added to the limitations section.

This study is quantitative and cross-sectional, meaning it has limits in showing cause-and-effect relationships. The self-reported data could also be influenced by social desirability bias, where participants may answer in a way they think looks good. Future studies could use qualitative or mixed methods to better understand the barriers to telehealth use in Iran.

2. There is no mention on non-response bias, there were over 200 participants and only 160 responded to the survey, did the authors assess why the others didnot respond.

Response: Despite reminders to complete the distributed questionnaires, we did not investigate the reason for physicians' non-participation, because from a research perspective, participants were allowed to opt out of the study. Response: We agree with the reviewer and accordingly we add a limitation in this regard in the limitation section as follow:

Secondly, although most physicians participated in this study, we were unable to obtain responses and perceptions from some of them regarding their use of telehealth services. Therefore, our results may be affected by non-response bias.

---

## [Editor Report · Decision Letter 2]

Barriers to widespread adoption of telehealth from physicians’ perspective: A survey in southern Iran

PONE-D-24-58379R2

Dear Dr. Alipour,

We’re pleased to inform you that your manuscript has been judged scientifically suitable for publication and will be formally accepted for publication once it meets all outstanding technical requirements.

Kind regards,

Marsa Gholamzadeh, PhD

Academic Editor

PLOS ONE

Additional Editor Comments (optional):

Thank you for your attention to detail in responding to comments. All comments have been well answered and the bugs have been fixed.
---

## [Editor Report · Acceptance letter]

PONE-D-24-58379R2

PLOS ONE

Dear Dr. Alipour,

I'm pleased to inform you that your manuscript has been deemed suitable for publication in PLOS ONE. Congratulations! Your manuscript is now being handed over to our production team.

Kind regards,

on behalf of

Dr. Marsa Gholamzadeh

Academic Editor

PLOS ONE